# Factors Associated with High Mercury Levels in Women and Girls from The Mojana Region, Colombia, 2013–2015

**DOI:** 10.3390/ijerph17061827

**Published:** 2020-03-11

**Authors:** Sonia Mireya Diaz, Ruth Marien Palma, Maria Nathalia Muñoz, Carolina Becerra-Arias, Julián Alfredo Fernández Niño

**Affiliations:** 1Group of Environmental Risk Factors, National Institute of Health, Bogotá 111321, Colombia; sdiaz21@gmail.com (S.M.D.); natamunozg@gmail.com (M.N.M.); 2Environmental and Labor Health Group, National Institute of Health, Bogotá 111321, Colombia; rpalma@ins.gov.co; 3Secretary of Health and Environment of Bucaramanga. Public Health Surveillance, Bucaramanga 680006, Colombia; caroba23@hotmail.com; 4Department of Public Health. Universidad del Norte, Barranquilla 081007, Colombia

**Keywords:** mercury, metal, exposure, women, fish

## Abstract

Women are primarily exposed to mercury through the consumption of fish contaminated by gold mining activities. The main systems affected are the central nervous and renal systems, although effects on the reproductive system have also been found. *Objective*: To explore the relationship between mercury levels in women and their possible sources of contamination. A cross-sectional study was conducted from 2013 to 2015 with women residing in 11 municipalities in La Mojana, Colombia, using non-random sampling. Standardized instruments were used to identify sociodemographic characteristics, mercury use, mining-related activities, water and fish consumption, and other factors. Blood, urine, and hair samples were taken to quantify mercury levels. A logistic regression model was used to identify factors associated with elevated mercury values in the participants’ samples. A total of 428 women were included, with an average age of 36.7 ± 16.7 years, 3.3% of whom were pregnant at the time of the survey and 8.4% of whom were exposed occupationally. High levels of mercury were present in 62.8% of the women, in any one of the three samples processed. Those exposed occupationally and environmentally had similar values (*p* = 0.821). Frequency of fish consumption and source of drinking water were associated with higher levels of mercury (*p* < 0.05).

## 1. Introduction

Gold production has increased in Colombia in recent years, especially artisanal and small-scale gold mining (ASGM) in areas that have traditionally been used for this purpose. Unfortunately, some mining activities lack the environmental licenses that certify that these practices are suitable. In Colombia, gold can be extracted in two ways. One way is through underground mining, in which the gold is confined to veins beneath the surface. Another method is called placer mining, in which gold is found in alluvial deposits. In this case, gold flows from an original source, such as a vein, until it reaches parts of the river where the water flow slows. Since these methods use large amounts of mercury, both of these extraction techniques pose a risk to human health and the environment. During these processes, large amounts of mercury are released into the environment, which then reach nearby fresh water bodies where the mercury can be methylated and subsequently bioaccumulated in aquatic fauna. Due to its density, the gold accumulates at the base of placer deposits [1]. 

The majority of mining activities in Colombia are carried out by placer mining (51%) and underground mining (30%) [1]. Hg appears in the environment in three general forms: elemental, inorganic, and organic. Humans are primarily exposed through the consumption of fish that have high levels of methylmercury (MeHg), an organic form of Hg that is the most toxic [2,3,4,5,6]. While the central nervous and renal systems are the main systems that are affected by mercury exposure, effects on the female reproductive system have also been found [7,8].

Women are most vulnerable to exposure during the preconception, pregnancy, and postnatal periods, given that MeHg crosses the placental barrier. This can have different effects depending on the dose. Findings have shown that children who are exposed prenatally to high levels of MeHg can be affected in various ways, including reduced brain size; cortical blindness; motor deficiencies; impaired auditory function, language development and memory; low IQ; impaired visual-spatial abilities; and mental retardation [8,9,10,11]. These effects may become evident over the medium- or long-term and are potentially irreversible [12]. Although reduced IQ is not easily detectable at the individual level, it can be identified at the population level when assaying exposed communities [13]. Some studies have shown effects on both human and animal reproduction after exposure to mercury, including an increase in miscarriages with high levels of mercury in the urine. Other findings include associations with stillbirths, birth defects, and menstrual disorders (irregular and painful periods and bleeding) [8,14,15].

Intrauterine exposure to mercury is typically monitored with biological samples, including umbilical cord blood, breast milk, and children’s hair [16,17]. However, in order to make proper public health decisions, mercury levels should be monitored before conception in women of childbearing age who reside in the most vulnerable regions and the sources of exposure should be identified. The present study explored the relationship between mercury levels in three biological matrices (blood, urine, and hair) and their possible sources of contamination among women in Colombia’s Mojana region from 2013 to 2015.

## 2. Materials and Methods 

### 2.1. Design and Study Site

This cross-sectional study was carried out between 2013 and 2015 with women who were residing in 11 municipalities in the La Mojana region of Colombia: Tiquisio, San Martin de Loba, Arenal, Guaranda, Majagual, San Marcos, Buriticá, Caucasia, El Bagre, Ayapel, and Montelibano. These municipalities were selected because their populations mainly work in artisanal gold extraction. They are located in four of Colombia’s departments: Bolívar, Sucre, Antioquia, and Córdoba. 

### 2.2. Test and Selection Criteria 

Non-probabilistic sampling was used to select a total of 428 women from the municipalities studied. To this end, meetings were held before conducting the study in order to explain the subject and objectives of the research as well as all of the procedures in which the participants would be involved. After they were informed, agreement to participate in the study was obtained. In order to inform the entire community about the study to be performed, messages were sent to neighborhoods through local media with the support of municipal health authorities. The participating women agreed to voluntarily take part in the study and signed informed consent. Informed consent for children under 18 years old was given by their parents or guardians. The inclusion criteria were: being residents of any of the municipalities studied, having lived in the area for at least six months, and if engaged in mining activity, having done so for at least six months. Exclusion criteria were women who reported having a neurological or cerebrovascular disease or a mental disorder, such as schizophrenia or bipolar disorder. 

### 2.3. Data Collection Instruments and Procedures

A questionnaire was used, which was adapted from the Global Mercury Assessment [18]. It included socio-demographic data, eating habits, water and fish consumption, and mining activities. Study participants were also asked about the presence of signs and symptoms compatible with mercury poisoning, cigarette and alcohol consumption, and exposure to other toxic substances over the last year. The questionnaire was administered by health professionals who did not know the study participants or their exposure levels and who were trained in the use of the instrument. Additional details about the methodology are found in a previous publication [19]. 

### 2.4. Quantification of Mercury 

Samples of venous blood (10 mL), urine (50 mL), and hair from the occipital region of the scalp (10 mg) were taken from each participant. These were analyzed with Direct Mercury Analyzer (DMA-80) TriCell Milestone equipment according to the U.S. Environmental Protection Agency (EPA) 7473 method (thermal decomposition, amalgamation, and atomic absorption spectroscopy), using cold vapor atomic absorption spectroscopy (CVAAS) [20]. The method’s detection limit was 0.87 g/L and the quantification limit was 2.6 g/L. The samples were processed in the laboratory of the Environmental and Labor Health Department of the National Institute of Health (NIH). The NIH uses an internal laboratory quality system in order to obtain accurate results for each of the measurement procedures. It also participates in external quality programs with the Centre de Toxicologie du Québec (Canada) for heavy metal measurements. The standard concentrations are between 0.1 to 100 ug/L for blood and urine and from 100 to 700 ug/L for hair. The proportion of mercury recovery was 90% to 100%. The presented units for the results are ng/dL, equivalent to µg/L. 

### 2.5. Case Definition 

With regard to the analysis of the samples, values >5 µg/L in blood, >7 µg/L in urine, and >1 µg/g in hair were considered indicators of toxicity in environmentally-exposed women. Values >15 µg/L in blood, >25 µg/L in urine, and >2 µg/g in hair indicated high levels of occupational exposure. 

### 2.6. Statistical Analysis 

Qualitative variables were presented as proportions, with their respective 95% confidence intervals (CI). Quantitative variables were described using measures of central tendency (mean and median) and dispersion (standard deviation (±) and interquartile range (IR)). These variables were analyzed using histograms and p-norm and q-norm graphs. For continuous symmetric variables, the Student’s *t*-test was used to evaluate differences among groups. The Mann–Whitney test was used for non-symmetric variables. And the relationships between qualitative variables were explored using the chi-square method. With regard to multiple regression models, since the level of mercury was the response variable (dichotomous), the analysis was conducted using logistic regression, with exposure source as the main independent variable. Age, source of drinking water, schooling, and occupation were the main adjustment variables. 

For all of the regression models, graphical and numerical methods were used to verify all assumptions. The diagnostic of the final fitted model took into account the value of the pseudo R-squared. Additionally, the impact of extreme values was examined and eliminated by analyzing potentially influential or extreme values. All relationships with a *p*-value < 0.05 were considered statistically significant. The analyses were performed with Stata 12 software (Stata Corp., College Station, TX, USA). 

All women who were included agreed to participate and signed an informed consent prior to beginning the investigation. Participants who were found to have high levels of mercury in any of the biological matrices received their results individually from the municipal health authority, with the recommendation that they consult with their health organization. This study was approved by the Research Ethics Committee of the University of the Andes, Colombia (approval number 459/2015).

## 3. Results

### 3.1. Sociodemographic and Occupational Characteristics

The 428 participants were from 11 municipalities located in the four departments evaluated were divided as follows: Bolivar 26.6% (114), Sucre 19.2% (82), Antioquia 26.9% (115), and Córdoba 27.3% (117). The average age was 36.7 ± 16.7 years. Of the total, 132 (30.9%) were enrolled in Colombia’s contributory health system (health system paid by each individual), 281 (65.8%) belonged to Colombia’s government-subsidized health system, and 14 (3.3%) were not enrolled in the health system. Regarding education level, 41.7% (178) of the women had attended primary school, 29.7% (127) had attended secondary school, and 27.8% (119) had completed university or postgraduate studies. Of all the women, 63.1% (270) were in the childbearing age group (15 to 49 years of age). At the time of the study, 3.3% (14) were pregnant, with an age range between 13 and 41 years. 

With regard to occupation, of the 8.4% (36) women who were classified as exposed, 51.4% (18) had worked in mining for six years or less and the remaining 48.6% (17) had worked in mining from 8 to 37 years. Of all these participants, 66.7% (24) reported using personal protection elements when performing their work activities, including aprons, gloves, masks, respirators, hats, visors, and goggles. The most common mining jobs conducted by the women were use of a water source (sifting sand through a pan designed for this purpose) in order to extract gold particles (known as panning) (3.2%, *n* = 14), administrative tasks (1.4%, *n* = 6), and general services (1.17%, *n* = 5). Additionally, 3.8% (16) said they had worked in burning amalgam at some point in their lives and 2.4% (10) said they or a family member burned amalgam inside their home. 

Of the total study population, 95.6% (408) mentioned that they consumed fish at least once a month. Some of the most commonly consumed species were bream (69.4%), catfish (22.9%), tilapia (18.7%), and “bocachico” (13.6%). Appendix A lists additional factors that were evaluated.

### 3.2. Mercury Levels in Study Participants

Median mercury levels of those presenting occupational exposure were 6.9 (IR = 2.1–13.6) in blood, 14.0 (IR = 5.2–26.5) in urine, and 2.3 (IR = 1.2–3.2) in hair. For environmental exposure, those same values were 3.1 (IR = 1.5–7.3), 5.6 (IR = 4.1–9.6), and 1.2 (IR = 0.6–2.1). The above differences in mercury medians were statistically significant (*p* = 0.028). When comparing these values according to enrollment in the government health system, the median blood level for the unaffiliated group, 11.7 (IR = 4.1–25.9), was greater than those of the other groups. With regard to urine, the contributory health system had the highest value (median 8.9, IR = 5.0–26.5). The value for hair was highest for those who were not affiliated with the health system (median 2.2, IR = 1.4–6.7). In the population studied, 62.8% (269) presented values above the established maximum limit in any one of the three samples analyzed. The percentages of people with values over the maximum established limit were not statistically different for occupational versus environmental exposure. Nevertheless, these percentages were higher for those who consumed fish more than once per month or 2–4 times/week (*p* = 0.002), with higher median mercury levels for those participants (29.5% in each case) than for participants who consumed it less frequently. Appendix A presents the median mercury levels for each matrix according to exposure group.

In the bivariate analyses, the relationship between exposure and having high mercury levels was not significantly correlated (*p* = 0.821) for any of the three matrices. That is, levels of mercury in blood, urine, or hair were the same for women who were occupationally exposed and for those environmentally exposed. The correlation between mercury levels and age was also insignificant (*p* = 0.782). Regarding education, the prevalence of elevated levels for those with a university education or postgraduate degree was 50% less than those who had primary school or less education (*p* = 0.013). In addition, those not affiliated with any health system presented a 9-fold greater prevalence of mercury levels in the body (*p* = 0.034). Women residing in Córdoba showed higher levels than those in the other three departments (*p* < 0.001). With regard to the source of drinking water, drinking from rivers and wells increased the prevalence of high mercury levels by 2.3 times, compared to drinking from bottles (*p* = 0.006 and 0.001, respectively). In terms of fish consumption, all frequencies were associated with high levels: once/month (*p* = 0.045), once/week (*p* = 0.009), 2–4 times/week (*p* = 0.005), and daily (*p* < 0.001). On the other hand, none of the symptoms that were evaluated were found to be significantly correlated with an increase in these levels in any of the biological matrices. In relation to being pregnant at the time of the survey, pregnant women had higher levels of mercury in all three matrices, with a median of 6.7 (IR = 5.2–11.7) in blood, 6.3 (IR = 3.7–16.9) in urine, and 1.3 (IR = 0.9–1.9) in hair. While blood mercury levels were significantly higher in this group of women than in non-pregnant women (*p* = 0.037), these two groups had similar mercury levels in urine and hair (urine *p* = 0.750 and hair *p* = 0.398).

When disaggregating the mercury levels of the participants according to frequency of fish consumption, the values in blood (*p* = 0.482) and urine (*p* = 0.462) were found to be similar, while the values for hair were not (*p* < 0.001) (Figure 1). That is, the more frequent the fish consumption among the women in Mojana, the higher the mercury values in hair, with a statistically significant difference (Figure 1d). And while median values in blood increased with the frequency of consumption, the differences were not significant for this matrix.

### 3.3. Association between Exposure Factors and Mercury Levels 

In the bivariate analysis, high mercury values (in blood, urine, or hair) were significantly correlated with the consumption of river water (*p* = 0.006) and well water (*p* = 0.001) and with subjects having technical, university, or postgraduate studies (*p* = 0.013). With regard to the work trades, being a nursing assistant could indicate a higher prevalence of alterations in mercury values in the organism (*p* = 0.031). In the case of blood mercury levels, university/postgraduate studies (*p* = 0.043) were associated with values that were higher than the maximum permissible limit for this matrix. Secondary education (*p* = 0.018) was associated with mercury in urine. Consumption of river water (*p* = 0.001) and well and river water (*p* = 0.001), higher educational level (*p* = 0.001), fish consumption (*p* < 0.001), and the nursing assistant (*p* = 0.019) were correlated with a greater prevalence of high levels in hair. 

The multiple logistic regression models showed that the source of water consumption and frequency of fish consumption were associated with high levels of mercury, as compared with the maximum permissible limits established for each of the three matrices (Appendix A). The consumption of well water was especially associated with a high prevalence of mercury, which was three-times more than that of women who consumed tap water. When evaluating the values defined for each sample as a response variable, none of the covariates were found to be associated with high values in blood and urine. However, for fish consumption, women with a daily intake had a 33-times higher prevalence of mercury levels in hair compared to those who never ate fish. A similar result was found for consuming fish once/month, once/week, and 2–4 times/week, though to a lesser extent, as seen in Appendix A. 

## 4. Discussion

### 4.1. Artisanal Small-Scale Gold Mining (ASGM)

When conducting ASGM activities, various methods are used to extract several different minerals. One of those minerals is gold, which has seen rising prices in recent years, along with a consequent increase in gold mining activities. Annual gold production in the world is roughly 350–400 tons, which results in 1400 tons of mercury being released on land and in water and air [21]. In addition, ASGM is a common occupation in some of Colombia’s departments, such as Antioquia, Bolivar, and Sucre. Informal mining is common and is directly associated with illegal mining along with the failure of mining formalization programs and a lack of control over the importation of mercury [22].

### 4.2. Occupational and Environmental Exposure to Mercury

In the present study, 8.4% of the participating women presented occupational exposure, with an average mining activity duration of 10.4 years. Occupational exposure is mainly caused by mercury vapor, which could produce chest pain, dyspnea, cough, haemoptysis, impaired pulmonary function, and interstitial pneumonitis [23]. It is likely that the majority of participants in this study had been exposed for a prolonged period of time. In addition, the multivariate regression models found that water source and frequency of fish consumption were associated with high levels of mercury. This may suggest that these women could have been or could have become pregnant during the entire exposure period, in which case the mercury levels would present a threat for both themselves and their offspring. 

Non-occupational exposure to mercury was strongly related with environmental pollution. This type of exposure can result from the consumption of contaminated fish and water, especially when a population permanently lives in areas that are influenced by mining, as shown by Junaidi et al. in Indonesia [24]. Thus, exposure is likely to be exacerbated by the long average residence time for those living in the Mojana area, which in the present study was 19.6 years.

### 4.3. Blood Mercury Levels after Exposure

With respect to mercury levels in the participants, 62.9% presented values in excess of the cutoff point defined for each sample (95% CI = 58.2–67.3). The estimates obtained were 32.4% (95% CI = 27.7–37.5) for blood, 36.2% for urine (95% CI = 27.9–45.5), and 56.9% for hair (95% CI = 52.2–61.6). The literature has shown that women who consume more fish generally have elevated blood mercury levels [25,26,27]. It has been shown that mercury levels are also related to socioeconomic level, where those in higher levels may consume more fish in accordance with recommendations to consume more omega-3 (mainly from this source) in order to maintain a healthier diet. Therefore, the literature has concluded that women of reproductive age who live in coastal areas (mainly the Atlantic and Pacific coasts) consume fish more frequently and have higher blood mercury levels [25,26]. This may be the case in the present study, given that the study population lives in coastal cities where fish is one of the main sources of dietary intake. In addition, the concentration of mercury in hair has been reported to be the most useful indicator of mercury exposure in both children and adults. This is because the concentration of methylmercury in hair is proportional to the blood concentration at the time when the hair strands form, such that the measured concentration in hair would be 250-times the blood mercury concentration [27,28]. This study found that the frequency of fish consumption was associated exclusively with mercury levels in hair, which suggests that fish consumption is a good proxy for mercury exposure in this region. This was also evidenced by Costa Junior et al. [29] in their study of the Pará region of Brazil and Santos et al. in their research on an indigenous population in that country [30].

### 4.4. Fish Consumption, High Mercury Levels, and Effects on Fertility

In developing countries, such as Colombia, fish consumption and contaminated water have been determined to be the main sources of environmental exposure to mercury [24]. However, fish is an important source of nutrients, such as polyunsaturated fatty acids, choline, selenium, vitamin D, and docosahexaenoic acid (DHA), which are all essential during various stages of life [31]. Oken et al. found high mercury concentrations in the blood of pregnant women whose children had neurodevelopmental alterations by the time they reached 3 years of age [32]. Such levels are related with fish consumption during pregnancy [26]. Since 2001, the United States Environmental Protection Agency (US EPA) has recommended that pregnant women avoid eating fish with high concentrations of mercury or limit total fish intake to no more than two 6-ounce servings (170 grams) per week [33].

In 2017, Figueroa et al. reported that frequent fish consumption was associated with lower socioeconomic status and being afro-descendant [6]. A general finding from that study was that 70% of the women studied who consumed a fish species called “manteco” at least once a week were afro-descendant. This stemmed from the cultural pattern of this ethnic group, which is related to the geographic location of these women. These factors are linked with high levels of heavy metals, including cadmium and mercury [6]. The authors of another study found that prior exposure to mercury vapors from mining activity was associated with irregularities in the menstrual cycle of Colombian women, but not with miscarriage [7]. 

Of the total participants in the present study, 3.3% were pregnant at the time it was conducted. Of these, 64.3% had high levels in any one of the three matrices analyzed. Specifically, we identified statistically significant differences in blood mercury levels (median 6.7; IR = 5.2–11.7) between pregnant and non-pregnant women. The consequences of such exposure would affect neurodevelopment and neurocognitive, psychomotor, and mental development, as well as the behavior of exposed pregnant women [32,34,35,36]. 

In addition to the above, the ability of mercury to cross the blood-brain and placental barriers represents potential exposure to the baby [8]. A study of pregnant women conducted in Hawaii between 2010 and 2011 found that the consumption of more than 12 ounces of fish per week was significantly associated with higher levels of mercury in umbilical cord blood after birth. Moreover, consumption was related to the income and ethnicity of the women [10]. 

The latter point is relevant to the findings of the present study, given that the cultural component that is common to the study region could influence the consumption habits of the population, such as a higher consumption of river water and fish. Pregnant women in the study reported consuming different types of fish, including bream, tilapia, and catfish, species that have been reported to have elevated levels of Hg in contaminated areas. This is of concern for both the health of the mothers as well as their developing fetuses. 

### 4.5. Variables for Evaluating Associations with High Mercury Levels

One of the limitations of the present study was not having evaluated some of the variables that could be related with mercury increases in the women studied. Examples include dietary factors other than fish consumption, socioeconomic level, race, and nutritional status [6]. Regarding additional dietary information, the consumption of other foods and nutrients is an aspect to consider when assessing mercury, especially fruits, vegetables, grains, and omega-3. Some of those foods (e.g., Brazil nuts) have been reported to have a mediating effect on mercury in the human body [37]. 

Studies have found that mercury values are related to the frequency of fish consumption as well as the type and source of fish (river, lake, sea) [37,38]. However, the present work did not have information on the location of the source of the fish consumed or on housing or mining activity. This study should be understood based on the culture and context that are common in areas where small-scale mining is practiced. Such is the case of the Colombian Mojana region, where the populations nearest to the center of such activity may present higher levels than more distant areas or non-mining regions [37]. Furthermore, in addition to exposure from ingestion, the evaluation of other routes that may also contribute to mercury levels in this population is pertinent, such as inhalation [38]. Similarly, because there was no control group (from non-mining areas, for example), it was not possible to determine differences in mercury levels according to area of residence or differences with areas in the Mojana region where everyday activities were different. 

Another limitation of the results herein involved the self-reporting of symptoms that may or may not be related with mercury toxicity. This was due to the methods we used to collect information, in which we inquired about signs or symptoms that the women identified, as applicable. Although the information constitutes a relevant baseline for our analyses, it was not possible to rule out overestimation or underestimation relative to additional factors that are associated with mercury exposure or to weak or undetected factors. Considering the above, the prevalence of the symptoms presented must be interpreted cautiously. They could be taken together with the principal findings in order to generate hypotheses about the factors and symptoms associated with mercury exposure, which could then be evaluated using designs that enable assessing that conclusion.

### 4.6. Applicability of the Findings 

In terms of the strengths of this work, we should mention the use of the population approach, which included all women in the Mojana region from 2013 to 2015. Accordingly, the results could be generalizable to that area and others with characteristics similar to those studied. Furthermore, while other studies have explored similar aspects in the country, they used descriptive methods [20,23,39,40]. Therefore, the present work is one of the first to use an analytical focus in order to identify factors associated with high mercury values in a mining population. This can serve as a basis for new research questions that can be resolved with new techniques to elucidate factors related with mercury toxicity in areas where mining is one of the main activities. 

Standard procedures were used to collect the information and analyze the biological samples. Procedures by the National Institute of Health of Colombia were used for the latter. This helped to minimize potential information bias due to systematic errors that can be generated by using non-parameterized techniques. In addition, prior education and training of the staff contributed to obtaining valid and generalizable results.

Lastly, although the present study did not analyze the impacts of mercury levels on fetuses or newborns, the prevalence of associated factors and the high concentration that was found in the biological matrices suggest that these effects may exist in this population of women and children, and that they could continue over time. This should be studied by future investigations and should serve to alert health authorities. 

## 5. Conclusions

The present findings indicated that, for Colombian Mojana women, the frequency of fish consumption and source of drinking water are associated with higher levels of mercury in the matrices evaluated. Therefore, fish consumption would be a predictor of mercury values found in the body. Given that a large amount of gold mining is conducted in this area, the importance of this conclusion is that both environmental as well as occupational exposure would contribute to the above assertions. Considering that mercury has a potential strong effect on the health of women and their offspring, it is important to public health to identify the sources of environmental exposure for which intervention is possible, especially for vulnerable populations, such as those in Colombia’s Mojana region. 

## Figures and Tables

**Figure 1 ijerph-17-01827-f001:**
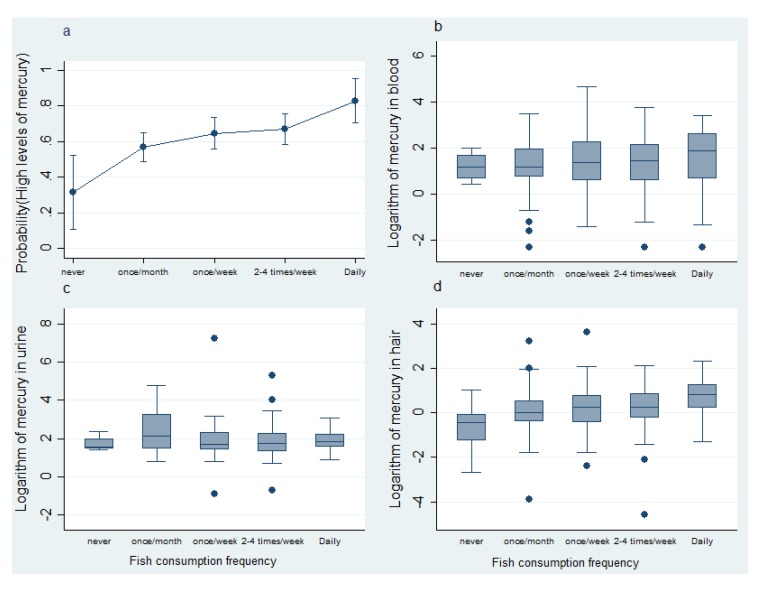
(**a**) Probability of high levels of mercury in at least one of the matrices evaluated. (**b**) Box diagram for blood mercury levels according to frequency of fish consumption. (**c**) Box diagram for mercury levels in urine according to frequency of fish consumption. (**d**) Box diagram for mercury levels in hair according to frequency of fish consumption.

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
