# Peer review of "Factors Associated with High Mercury Levels in Women and Girls from The Mojana Region, Colombia, 2013–2015"

_ijerph, 2020, doi:10.3390/ijerph17061827_

Round 1
Reviewer 1 Report
I think the information in here is important, but the way it is presented, is poor and difficult to read. Transitions could be improved. I have added some suggestions in my comments to assist in this; additionally, I would recommend an English edit.
Abstract
Abstract needs to indicate that the origin of the source is from ASGM.
Introduction
Why just list women? Is the main form of the exposure different for other genders/ages? What is the link between mercury as a chelating agent and women consuming fish? Maybe you should focus on the fact that mercury contributes to environmental pollution as well and can be harmful to aquatic life, which affects us through bioaccumulation.
The first paragraph needs to highlight ASGM, Colombia, routes of exposures, gender-based differences, and health inequality. These, as of right now, are not mentioned and/or not in that order.
Discussion
“The impact on public health…” Rewrite and say what it is.
First discuss mining, then mercury, then health effects, and then your study results and how it links in and/or supports.
Why is it high in fish? Explain. And why are people drinking contaminated water?
Figueroa and colleagues (year)…
Cite sentence “It has been demonstrated…” By whom?
Did you just assume that the women were not working in mining and had no exposure to it? I would state this as a major limitation.
Conclusion
Needs to follow this pattern: Colombia suffers from the most mercury contamination via ASGM. This translates to population health being affected. Mercury can contaminate and cause adverse health effects via direct inhalation, ingestion, and consumption of food (e.g. fish) or water. In one region, there is more fish consumption, or it is a staple so it is important to address mercury combination and exposure. This study focused on women WHY. Ultimately, it was determined that…
Author Response
Reviewer comment |
Adjustment done |
Answered in |
Abstract needs to indicate that the origin of the source is from ASGM |
It was indicated that contaminated fish consumed by women was a product from gold mining in the Mojana región. |
Line 20 |
Why just list women? Is the main form of the exposure different for other genders/ages? What is the link between mercury as a chelating agent and women consuming fish? Maybe you should focus on the fact that mercury contributes to environmental pollution as well and can be harmful to aquatic life, which affects us through bioaccumulation. |
We have a previous publication with all the population in la Mojana as the studied population (reference # 27). That is why we were encouraged to study mercury levels in girls and women, taking into account the effects of bioaccumulation of mercury on the reproductive system, as well as in the fetus for those pregnant ones that are exposed to mercury. |
|
The first paragraph needs to highlight ASGM, Colombia, routes of exposures, gender-based differences, and health inequality. These, as of right now, are not mentioned and/or not in that order. |
The first paragraph was re-written with the reviewer´s observation. Gender based are not mentioned as currently literature doesn´t show them, but the focus of the present job is on women because of the risk of neurodevelopmental deficits for fetus with mercury contamination through placental and breast milk transmission. https://link.springer.com/article/10.1007/s002440010195. |
Lines 39-49 |
“The impact on public health…” Rewrite and say what it is. |
Sentence was re-written to explain what the authors were trying to say |
Line 272 |
First discuss mining, then mercury, then health effects, and then your study results and how it links in and/or supports. |
The first part of discussion section was re-ordered to follow the reviewer comment |
Line 262-299 |
Why is it high in fish? Explain. And why are people drinking contaminated water? |
Some species as tilapia and catfish are reported for having high levels of mercury, as a contamination product during mining activities and the release to rivers. The contaminated water consumed by women and generally by most of the population in la mojana would result from mining activities in the zone, especially for those informal ones. That would be reflected on the association found between water form the well and high levels of mercury in the studied women |
|
Figueroa and colleagues (year)… |
The publication year was included in the text |
Line 347 |
Cite sentence “It has been demonstrated…” By whom? |
The authors were included in the sentence |
Line 358 |
Did you just assume that the women were not working in mining and had no exposure to it? I would state this as a major limitation. |
According to the occupation variable, women in the area could work in mining activities or not, despite this, they were considered as environmentally exposed while the ones working in gold mining were occupationally exposed. The absence of a control group (not exposed at all) is actually one of the limitations the authors discuss. |
Line 410 |
Needs to follow this pattern: Colombia suffers from the most mercury contamination via ASGM. This translates to population health being affected. Mercury can contaminate and cause adverse health effects via direct inhalation, ingestion, and consumption of food (e.g. fish) or water. In one region, there is more fish consumption, or it is a staple so it is important to address mercury combination and exposure. This study focused on women WHY. Ultimately, it was determined that… |
The discussion was adjusted to follow the structured suggested, an ASGM paragraph was included. |
Lines 261-313
|

Reviewer 2 Report
This is an interesting study that looks to characterise social science perpectives with documented physical science data in regards to Hg exposure to women of child-bearing age in small scale gold mining regions of Colombia. This novel approach does highlight the value of this study that may act as an avenue to explore these interactions in similar studies across contaminated areas in the future. I especially appreciated the authors attempts to discuss their own short-comings in their study. This is quite often passed over by scientists, but is important to place the study within the literature.
Nonetheless, this work need significant improvement before it can be published. In reviewing this study, I identified the following overarching issues that need improvement before I can recommend publication:
There is A LOT of over-elaboration. Whole paragraphs of things that could easily be summarised in one sentence. As a first-language English speaker that also speaks Spanish, I can understand where some of these problems come from: Spanish is a very "wordy" language. Nonetheless, this is not the only source and the authors need to realise this is mostly a physical science study and concise language is important. Drastically cut-down the meandering sentences and paragraphs that don't really go anywhere and are often repetitive please. The manuscript is very disjointed. Quite often I can reading a paragraph that was meant to be about one variable (or outcome) that included discussion on one or more other variables (or outcomes). Quite often one sentence had almost not relation to the following sentence. Additionally, the section titles were useful, but sections sometimes had repeated information or information that did not relate to the section it was in. The authors used a results section AND a discussion section. One of the problems that can often happen with this structure is that many data points and descriptions get repeated in the two sections. The authors need to limit the "discussion section" to discussion and not repeat results. There needs to be a much better quality assurance and control of the mercury measurements. How was the system calibrated. What standards were run? What were the recoveries of these standards? The form of data reporting is not clear at all. The description of what the "cutoffs" are and why those values were chosen is very vague and yet the whole study is based around those values. They need to be described in detail, not just referred to another study. Also why not discuss the actual concentrations measured? What were the maximum, what were the means, standard deviations for each group, not simply how many were above "cutoff" concentrations. Such data would be invaluable for the physical scientific community and MUST be included in the study. Abstract needs rewriting.This is definitely a lot of work, but I do feel this can be a useful contribution to the literature and that the authors put in the required effort to make the manuscript viable.
SPECIFIC COMMENTS:
Lines 39-39: Hg is used to form amalgams with gold, not chelates.
Lines 39-40: "Hg has three general forms..."
Line 41: "through the consumption of fish that have elevated levels of methyl-mercury, an organic form that is the most toxic
Lines 48-58: This section is all over the place and very disjointed. This is the 3rd reference to the nervous system. In between these references each time there is discussion on something else. Keep it simple...
Line 48: I don't think MeHg actively "targets" the brain (it isn't nefariously seeking out brains like a zombie)...I think a term less leading would be better. Also saying neurotoxin kind of makes the use of brain redundant.
Line 50: use MeHg instead of "this substance"
Lines 57-58: "A reduced IQ is not easily detectable at the individual level, but populationally when dealing with exposed inhabitants can be identified at the population level when assaying exposed communities." should be changed
Line 92: Here you have one reference yet state "previous publications". This should be singular.
Lines 148-162: This whole paragraph about fish exposure is all discussed again in the next section. Please avoid repetition and discuss fish related exposure in section 3.2
Lines 164-174: This was all discussed in the previous section. Why is it being repeated again?
TABLE 1 & TABLE 2: This are really just data reporting tables and don't add much value since all the interesting stuff is mentioned in the text. To make them worthwhile add actual mean and standard deviation concentration data of each population or move them to a supplementary information section and add a little more information in figures. And what is the "OR" Odds Ratio estimation? It isn't described anywhere. So what are the authors actually using it for? If nothing then why report it?
Lines 201-209: Here again the authors jump back and forth between fish, employment, education related relationships. Talk about the fish in one section only. Talk about occupational related exposure in one section only...etc.
Lines 224-225: The authors need references to confirm this latter point. It might seem "obvious" but it is an incredibly important point that most people just assume when it should be supported by literature.
Lines 228-229: Again, this is the same point I just highlighted (lines 224-225) and it is assumed again and not supported by the literature despite other points being so.
Lines 230-231: This is simply not true. Absolulely the most prevelant mechanisms for Hg exposure is through the consumption of fish. Consumption of fish is a non-occupational exposure. Yet, exposure to high levels of Hg vapour is generally much more related to occupational exposure. I.e. exposures to people working in gold shops and burning amalgams.
Lines 231-233: How does the consumption of contaminated fish cause exposure to Hg vapour? Which is EXACTLY what the authors are stating. Again this paragraph is very disjointed - described vapour exposures, fish consumption exposures and both non-occupational and occupational exposures. What are the authors actually trying to say?
Lines 233-235: what is an environmental illegality? Very ambiguous term that once again has no reference to the literature.
Lines 235-236: This last sentence should be deleted and the reference used in the previous sentence. Nonetheless, it is still too vague.
Lines 238-239: Residence time of the population or of Hg in the environment? Be careful here. Remember, residence time in chemistry means how long a contaminant persists in the environment.
Lines 239-241: Another very confusing and misdirected sentence and paragraph. Just state it simply. Multi-regression models show [contaminated drinking?] water source and increased fish consumption frequency had the greatest effect on human Hg exposure. Nonetheless, this is simply repeating what was written in the results. There is very little actual discussion here.
Lines 241-243: So the authors here are essentially saying: "Hg contacts water and transforms into methylmercury"...That could not be further from the truth. I hope the authors at least have a basic knowledge of how and where Hg gets methylated. And in mining areas there is much debate over this as only contamination in and around mining areas in waterways (sediments and water samples) are only elevated in a very very localised area.
See Moreno-Brush et al., 2019.
Lines 244-245: this is so unspecific.
Lines 250-256: Again all this could be written in one sentence:
"Literature has shown that women consuming more fish generally has elevated levels of blood mercury [31-32...here the authors could add more studies on this]."
Lines 256-259: Why is hair the best indicator? Just because someone stated it doesn't mean that it is fact. Discuss it with reference to the literature!
Lines 259-261: "The association between mercury levels in hair and the frequency of fish 259 consumption, found exclusively by this matrix in the present study, suggests that mercury levels are reflected by prolonged consumption that hair concentration is likely a good proxy for fish consumption." Should be changed.
Lines 261-263: "an" indigenous population, or indigenous population"s" or "the" indigenous populations. The latter would suggest across the whole country, which i am sure the referred to study did not do.
Lines 265-266: While this statement is much more factual, it is in complete contrast to what is written on lines 230-231.
Line 276 and Line 279: "...being black", this is NOT the correct terminology to use here. "Of African decent"...would be far more appropriate if the ethnicity of the individuals needs to be highlighted. This is a scientific study, not a colloquial discussion on the street. See: http://dx.doi.org/10.1136/jech.2003.013466
Lines 284-286: Who demonstrated this point? References please?
Lines 287-289: Who demonstrated this point? References please?
Lines 299-305: "In addition to the above, the ability of mercury to cross the blood-brain and placental barrier entails exposure to the baby, even greater than possible after birth [20]. In this sense, between 2010 and 2011 a study of pregnant women in Hawaii found that the consumption of more than 12 ounces of fish per week was significantly associated with higher levels of mercury in umbilical cord blood after birth, this included pregnant women [19]. In addition, pregnant women consuming seafood in this sample were more likely to have elevated levels of mercury than non-consumers. Moreover, consumption was related to the income and ethnicity of the women [19]." Use concise language. Should be changed.
Lines 306-308: I have no idea what the authors are trying to say here. And again "white" is not a term used in science to describe ethnicity.
Lines 309-310: This has already been discussed AT LENGTH, why are the authors repeating it again.
Lines 306-313: "Pregnant women in the study reported consuming different types of fish, including bream, tilapia and catfishAs is well known, chronic exposure to contaminated fish, even at low levels relative to the standard [44], is reflected by high levels of mercury in hair. This implies a greater vulnerability for this population group, not only because of its biological status per se but also because of the consumption of species with high levels of mercury, which could increase exposure for them and their babies, species that have been reported to have elevated levels of Hg in contaminated areas. This a concern for both the health of the mothers and their developing fetuses."
Use concise language. Should be changed.
Lines 317-319: This is pure hearsay unless it can be substantiated with data or in the literature and should be deleted.
Lines 321: "the consumption of other foods and nutrients..."
Lines 322-324: "what might have an mediating effect? again this is too ambiguous. If there is a study that states food X (let's say Brasil nuts for example...hint, hint!!) then state this specifically."
Lines 330-332: Ingestion and inhalation are two completely different exposure mechanisms. Inhalation is not a form of ingestion.
Lines 335-336: "probably that may or may not be related to mercury toxicity" The authors do not know that. Stating "probably" is very leading.
Lines 362-364: This is one place that doesn't need referencing. I would suggest removing the Diez reference here.
Lines 369-371: Who did the analysis?
Author Response
Reviewer comment |
Adjustment done |
Answered in |
Lines 39-39: Hg is used to form amalgams with gold, not chelates. |
The introduction was adjusted according to reviewers comments, that sentence in line 39 was eliminated |
Introduction |
Lines 39-40: "Hg has three general forms..." |
the sentences was adjusted |
Line 49 |
Line 41: "through the consumption of fish that have elevated levels of methyl-mercury, an organic form that is the most toxic |
the sentences was adjusted |
Lines 51-52 |
Lines 48-58: This section is all over the place and very disjointed. This is the 3rd reference to the nervous system. In between these references each time there is discussion on something else. Keep it simple... |
The paragraph was revised for simplification of the idea, it was also placed earlier in the text, and some sentences were deleted |
Lines 51-62 |
Line 48: I don't think MeHg actively "targets" the brain (it isn't nefariously seeking out brains like a zombie)...I think a term less leading would be better. Also saying neurotoxin kind of makes the use of brain redundant. |
The sentences was eliminated as it did not have complete relation with the above and below sentences |
|
Line 50: use MeHg instead of "this substance" |
The Word was chenged |
Line 56 |
Lines 57-58: "A reduced IQ is not easily detectable at the individual level, but populationally when dealing with exposed inhabitants can be identified at the population level when assaying exposed communities." should be changed |
The sentence was changed |
Lines 61-62 |
Line 92: Here you have one reference yet state "previous publications". This should be singular. |
It was stated as singular |
Line 111 |
Lines 148-162: This whole paragraph about fish exposure is all discussed again in the next section. Please avoid repetition and discuss fish related exposure in section 3.2 |
The paragraph was cut and put in the 3.2 section. |
Lines 183-198 |
Lines 164-174: This was all discussed in the previous section. Why is it being repeated again? |
The paragraph were re-organized |
Lines 158-198 |
TABLE 1 & TABLE 2: This are really just data reporting tables and don't add much value since all the interesting stuff is mentioned in the text. To make them worthwhile add actual mean and standard deviation concentration data of each population or move them to a supplementary information section and add a little more information in figures. And what is the "OR" Odds Ratio estimation? It isn't described anywhere. So what are the authors actually using it for? If nothing then why report it? |
Table 1 and 2 were moved as supplementary material and indicated in the manuscript. The OR is an estimation widely used in epidemiologic studies, it is stated as a footnote in table 2. It is presented in the results from multiple regression model as it is a measure to know how frequent it is the outcome in the exposed group compared to the non-exposed one. In the results sections it is stated “the consumption of well water was especially associated with a prevalence of high mercury, three times greater than that of women who consumed tap water” the three times refers to the OR= 3,6, as shown in table S2. For the associations found it was included in the results thee interpretation of hair levels according to OR. |
Lines 198 and 247
Lines 251-253
Lines 251-253 |
Lines 201-209: Here again the authors jump back and forth between fish, employment, education related relationships. Talk about the fish in one section only. Talk about occupational related exposure in one section only...etc. |
It was not possible to find the idea exposed as according to the submitted text lines 201 – 209 correspond to empty spaces before table 1, with no text |
|
Lines 224-225: The authors need references to confirm this latter point. It might seem "obvious" but it is an incredibly important point that most people just assume when it should be supported by literature |
Did you mean lines 270-271? two references were added to the sentence |
Line 273 |
Lines 228-229: Again, this is the same point I just highlighted (lines 224-225) and it is assumed again and not supported by the literature despite other points being so. |
Did you mean lines 275-276? two references were added to the sentence |
Line 276 |
Lines 230-231: This is simply not true. Absolulely the most prevelant mechanisms for Hg exposure is through the consumption of fish. Consumption of fish is a non-occupational exposure. Yet, exposure to high levels of Hg vapour is generally much more related to occupational exposure. I.e. exposures to people working in gold shops and burning amalgams. |
The paragraph was corrected to try to explain the non-occupational mercury expisition. |
Lines 294-299 |
Lines 231-233: How does the consumption of contaminated fish cause exposure to Hg vapour? Which is EXACTLY what the authors are stating. Again this paragraph is very disjointed - described vapour exposures, fish consumption exposures and both non-occupational and occupational exposures. What are the authors actually trying to say? |
The paragraph was corrected to try to explain the non-occupational mercury exposition. |
Lines 294-298 |
Lines 233-235: what is an environmental illegality? Very ambiguous term that once again has no reference to the literature. |
The paragraph containing the sentence was deleted taking into consideration the reviewer´s comments |
Line 302 |
Lines 235-236: This last sentence should be deleted and the reference used in the previous sentence. Nonetheless, it is still too vague. |
The sentences was deleted and the references placed in the previous one |
Line 330-331 |
Lines 238-239: Residence time of the population or of Hg in the environment? Be careful here. Remember, residence time in chemistry means how long a contaminant persists in the environment. |
We mean the inhabitants residence time in the Mojana, not the mercury in the environment as it was not measured. The Word inhabitants was included to make it clear. |
Line 298 |
Lines 239-241: Another very confusing and misdirected sentence and paragraph. Just state it simply. Multi-regression models show [contaminated drinking?] water source and increased fish consumption frequency had the greatest effect on human Hg exposure. Nonetheless, this is simply repeating what was written in the results. There is very little actual discussion here. |
The paragraph was corrected and simplified with an extra sentence for discussion |
Lines 285-290 |
Lines 241-243: So the authors here are essentially saying: "Hg contacts water and transforms into methylmercury"...That could not be further from the truth. I hope the authors at least have a basic knowledge of how and where Hg gets methylated. And in mining areas there is much debate over this as only contamination in and around mining areas in waterways (sediments and water samples) are only elevated in a very very localised area. See Moreno-Brush et al., 2019. |
Even this fact is widely known, in Colombia informal gold mining represents 87%, and that supposes a threat for the people working in this activity as well as for the environment, for the obvious reasons. Despite it is known, public health strategies and policies are needed to try to control this occupation, so the health effects of the exposed population could be diminished in middle and long time. That is why we consider studying this area could visualize the current situation, supported with designs and facts that make the results appropriate to take decisions at the political level. García et al published in 2014 the gold mining situation in Antioquia, which could support the hypothesis that the main mercury source to the environment in Colombia, in the known miner zones, would be from antrhopogenic emisions. |
|
Lines 244-245: this is so unspecific. |
The sentence was deleted |
Lines 292-293 |
Lines 250-256: Again all this could be written in one sentence: "Literature has shown that women consuming more fish generally has elevated levels of blood mercury [31-32...here the authors could add more studies on this]." |
The sentences was simplified as suggested, three references were cited in the sentence. |
Lines 318-321 |
Lines 256-259: Why is hair the best indicator? Just because someone stated it doesn't mean that it is fact. Discuss it with reference to the literature! |
An explanatory sentence was introduced with one more reference. |
Lines 327-331 |
Lines 259-261: "The association between mercury levels in hair and the frequency of fish 259 consumption, found exclusively by this matrix in the present study, suggests that mercury levels are reflected by prolonged consumption that hair concentration is likely a good proxy for fish consumption." Should be changed. |
It was changed |
Lines 332-332 |
Lines 261-263: "an" indigenous population, or indigenous population"s" or "the" indigenous populations. The latter would suggest across the whole country, which i am sure the referred to study did not do |
It was changed to “an indigenous population” |
Line 335 |
Lines 265-266: While this statement is much more factual, it is in complete contrast to what is written on lines 230-231 |
The paragraph in line 294-298 was adjusted in accordance to the sentence in line 337-338 |
Lines 294-298 |
Line 276 and Line 279: "...being black", this is NOT the correct terminology to use here. "Of African decent"...would be far more appropriate if the ethnicity of the individuals needs to be highlighted. This is a scientific study, not a colloquial discussion on the street. See: http://dx.doi.org/10.1136/jech.2003.013466 |
The term was changed to afro-descendant |
Linea 349-353 |
Lines 284-286: Who demonstrated this point? References please? |
Two more references were included in the paragraph, one of these in the sentence |
Lines 358-368 |
Lines 287-289: Who demonstrated this point? References please? |
Two more references were included in the paragraph, one of these in the sentence |
Lines 358-368 |
Lines 299-305: "In addition to the above, the ability of mercury to cross the blood-brain and placental barrier entails exposure to the baby, even greater than possible after birth [20]. In this sense, between 2010 and 2011 a study of pregnant women in Hawaii found that the consumption of more than 12 ounces of fish per week was significantly associated with higher levels of mercury in umbilical cord blood after birth, this included pregnant women [19]. In addition, pregnant women consuming seafood in this sample were more likely to have elevated levels of mercury than non-consumers. Moreover, consumption was related to the income and ethnicity of the women [19]." Use concise language. Should be changed |
|
Lines 375-381 |
Lines 306-308: I have no idea what the authors are trying to say here. And again "white" is not a term used in science to describe ethnicity. |
The idea was explicit in the sentence |
Lines 382-384 |
Lines 309-310: This has already been discussed AT LENGTH, why are the authors repeating it again |
It was deleted |
Lines 388-389 |
Lines 306-313: "Pregnant women in the study reported consuming different types of fish, including bream, tilapia and catfishAs is well known, chronic exposure to contaminated fish, even at low levels relative to the standard [44], is reflected by high levels of mercury in hair. This implies a greater vulnerability for this population group, not only because of its biological status per se but also because of the consumption of species with high levels of mercury, which could increase exposure for them and their babies, species that have been reported to have elevated levels of Hg in contaminated areas. This a concern for both the health of the mothers and their developing fetuses." Use concise language. Should be changed. |
The complete paragraph was revised with some adjustments. The suggestions were considered. |
Lines 382-387 |
Lines 317-319: This is pure hearsay unless it can be substantiated with data or in the literature and should be deleted. |
The sentence was deleted |
Lines 395-298 |
Lines 321: "the consumption of other foods and nutrients..." |
It was simplified |
Lines 399 |
Lines 322-324: "what might have an mediating effect? again this is too ambiguous. If there is a study that states food X (let's say Brasil nuts for example...hint, hint!!) then state this specifically." |
The cited reference (45) states the mediating effect of fruits, vegetables and omega-3 |
Line 402 |
Lines 330-332: Ingestion and inhalation are two completely different exposure mechanisms. Inhalation is not a form of ingestion. |
The mentioned limitation refers that in the present study it could not be determined whether the mercury levels in the studied population were the result of mercury inhalation, ingestion, or both. |
|
Lines 335-336: "probably that may or may not be related to mercury toxicity" The authors do not know that. Stating "probably" is very leading. |
“mayo r may not” was added |
Line 414 |
Lines 362-364: This is one place that doesn't need referencing. I would suggest removing the Diez reference here. |
The reference was removed |
Line 448 |
Lines 369-371: Who did the analysis? |
Supported was replaced by did |
Line 455 |

Reviewer 3 Report
Abstract
The aim in this study is not congruent with the title of this manuscript.
Introduction
Introduction
page 1 line 42-43
The authors of this citation did not find a statistically significant difference (p=0.067) and when calculating the prevalence ratio to explain the cause of mestrual irregularity, it was also not significant with a confidence interval between 0.93-2.73.
I suggest that the authors should cite an article that supports this relationship between the female reproductive system and mercury values in order to assume this relationship.
pag2 3 lines 44-47
The citation 13, only talks about problems of male reproduction and not about problems of female reproduction that would be the right thing to do according to the stated objective.
Page 3 lines 49-51
I consider that in this sentence the citation 20 (https://ipen.org/documents/mercury-women-child-bearing-age-25-countries) mentioned up to line 55 could fit perfectly.
Material & Methods
Page 3 line 98
This paragraph should be corrected, the units are wrong according to the manufacturer's brochure (Milestone).
Results
It would be recommended that in those data where you manage percentages you also include "n" participants.
According to your main objective you intend to find the relationship between mercury levels in women of reproductive age and their possible causes of contamination. However, on page 4 line 136-137 and table 1 you mention that only 63% of the participants were of childbearing age so your analysis must be modified because 37% of the participants analyzed are not of childbearing age and represent a bias in order to answer the research question.
Discussion
The authors do not discuss the possible relationship between mercury levels andt heir impact and importance in women of childbearing age
Conclusion
The authors do not answer the research question
Author Response
Reviewer comment |
Adjustment done |
Answered in |
The aim in this study is not congruent with the title of this manuscript. |
There was an extra Word in the objective: this study relates to women in general, of any age, at la Mojana región, not only for those at childbearing age. |
Lines 23 and 83 |
page 1 line 42-43 The authors of this citation did not find a statistically significant difference (p=0.067) and when calculating the prevalence ratio to explain the cause of mestrual irregularity, it was also not significant with a confidence interval between 0.93-2.73. I suggest that the authors should cite an article that supports this relationship between the female reproductive system and mercury values in order to assume this relationship. |
This is an important reference for the present job as it was developed in colombian mining population, despite the “almost singificant” findings. This could partly be explained because of the small simple size that authors mentioned could not be completed. A reference was added with results regarding child-birth defects as mercury exposure during pregnancy. |
Line 54 |
pag2 3 lines 44-47 The citation 13, only talks about problems of male reproduction and not about problems of female reproduction that would be the right thing to do according to the stated objective. |
In that sentence are included reference 11 and 12, which relate to effects of mercury exposure during pregnancy. Reference 20 was also included and 13 was deleted |
Line 66 |
Page 3 lines 49-51 I consider that in this sentence the citation 20 (https://ipen.org/documents/mercury-women-child-bearing-age-25-countries) mentioned up to line 55 could fit perfectly. |
The paragraph was deleted, as a suggestion from one of the reviewers |
Lines 67-77 |
Page 3 line 98 This paragraph should be corrected, the units are wrong according to the manufacturer's brochure (Milestone). |
Analysis characteristics for mercury measurment can be adapted in order to consider the evaluated population. In this case, eventhough the brochure indicates units as ng/g the analyser can determine the sample preparation method for processing. As mentioned in methodology with CVAAS the units are µg, as the method was standrized at NHI. Furthermore, according to WHO guidelines to identify mercury exposure in at risk populations, the suggested reference levels for the three biomarkers (blood, urine and hair) correspond to µg/L and ug/g. WHO. Guidance for identifying populations at risk from mercury exposure. The NHI Works with a within-laboratory quality system for having accurate results in each of the measurements procedures. Furthermore, it participates in external quality programs with the Toxycologic centre in Canada for heavy metals measurements. This system was calibrated with a curve using certified standards from ACCUSTANDARD. The presented units for the results are equivalent to ng/dL, as µg/L). |
Line 118 |
It would be recommended that in those data where you manage percentages you also include "n" participants. |
We included n for the percentages indicated in paragraphs 1 and 2 in the results sections that were missing.
|
Lines 149-178 |
According to your main objective you intend to find the relationship between mercury levels in women of reproductive age and their possible causes of contamination. However, on page 4 line 136-137 and table 1 you mention that only 63% of the participants were of childbearing age so your analysis must be modified because 37% of the participants analyzed are not of childbearing age and represent a bias in order to answer the research question. |
We made a correction in the stated objective in the manuscript, as initially we pretended to do it on childbearing age women but we agreed before writing the manuscript to include all women and girls of any age at the zone. |
Lines 23 and 83 |
The authors do not discuss the possible relationship between mercury levels andt heir impact and importance in women of childbearing age |
As mentioned in the previous comment the scope of the article is women of any age. Because of the study design it is not possible for the authors to conclude that high levels of mercury would be related to pregnancy or reproductive outcomes, as those variables were not measured for the study. In the introduction we cited some references that suggest a possible relationship between high mercury levels and reproductive outcomes, so this should be further studied. Despite this, the present results are the basis for further investigations that lead to a deeper conclusion. |
Lines 63-66 Lines 78-84 |
The authors do not answer the research question |
With the adjusted objective for the studied population, the research question ¿which are the associated factors to high mercury levels in women from la Mojana during 2013-2015? Is answered with the main results, stated In the conclusions section. |
Lines 444-452 |

Round 2
Reviewer 2 Report
Specific comments:
Lines 43-45: Please change this sentence to:
"where it CAN be methylated."
The authors are still confusing what can happen with what will happen all the time. Not all mercury is methylated and again the authors are showing a lack of understanding of how methylation occurs which is crucial to this whole study. Do the authors know methylation takes place almost predominantly in sediments, and predominantly in more still water bodies?
This sentence also needs a full-stop at the end.
Line 57-58: There should be no paragraph end between these two sentences.
Section 2.4: Information on quality control is for all readers not just a single reviewer. Why is this not included here. Please add what you wrote in the response:
"The NHI Works with a within-laboratory quality system for having accurate results in each of the measurements procedures. Furthermore, it participates in external quality programs with the Toxycologic centre in Canada for heavy metals measurements. This system was calibrated with ine and from 100 to for hair. The recovery percentages were 90 to 100 %. The presented units for the results are equivalent to ng/dL, as μg/L)."
Section 2.6: The use of proportions is fine for the main discussion of this article. But physical scientists want and need data of actual concentrations. These data MUST be presented in at least an SI, there does seem to be some of these data in the results. Nonetheless, please add a table to the SI with the median and standard deviation for each group along with the porportion data. This MUST happen to improve the impact of this study and support science by contributing to the greater data pool on ASGM human health data.
Lines 137-139: This comparison doesn't make sense. How many older women completed primary school and how many younger women complete secondar school. Why highlight different things from different age groups?
Lines 152-159: Is this the actual concentration values I have been asking for? These data must have units associated with them or they mean very little. Again, I would recommend adding a table with all the concentration data in the SI. This supports open science and also will allow other scientists to potential use such data in for future studies, such as meta studies looking at combining data on concentations in women at all ASGM sites globally.
Section 4.1: The new section 4.1 is all background information and should be moved to the introduction. The current section 4.2 should be the start of the discussion.
Lines 231-240: This is again all repeated information from the introduction. Delete it. Get to the point of the discussion...
Line 237: "CAN UNDERGO METHYLATION!!!" Not all mercury gets methylated!!!
Line 247: What is "the earlier"...I have no idea. This is not an expression used to describe an earlier point. "The former" maybe?!
Lines 253-254: "On the other side" of what? Again not an English expression. Just say, "exposure is likely exacerbated by the long average residence time for inhabitants in the Mojana area..."
Lines 284-286: This contradicts what the authors discussed in the previous section that higher education (as assumed socioeconomic status) means women eat more fish...so which is it?!
Lines 292-298: I feel like this whole paragraph is just over explanation of the health concerns described in the introduction...again. Mercury is bad for pregnant womens health. We understand this, as the authors have described it at length!! Delete this paragraph and get onto describing how this relates to your data (the next paragraph).
Line 297: What is "metilmercury?"
Lines 301-304: Again we know these health effects!!! you've said it over and over and over again...too much repitition. Delete.
Lines 323-325: Not therefore...please write: "some of those foods (e.g. Brasil Nuts) have been reported to have a mediating effect on mercury in the human body."
Author Response
Reviewer #2
Reviewer comment |
Adjustment done |
Answered in |
Lines 43-45: Please change this sentence to: "where it CAN be methylated." |
It was changed according to suggestion |
Line 48 |
The authors are still confusing what can happen with what will happen all the time. Not all mercury is methylated and again the authors are showing a lack of understanding of how methylation occurs which is crucial to this whole study. Do the authors know methylation takes place almost predominantly in sediments, and predominantly in more still water bodies? This sentence also needs a full-stop at the end. |
A sentence was included to try to clarify what the reviewer suggested |
Lines 49-51 |
Line 57-58: There should be no paragraph end between these two sentences. |
The sentence were joint in the same paragraph |
Line 64 |
Section 2.4: Information on quality control is for all readers not just a single reviewer. Why is this not included here. Please add what you wrote in the response |
A sentence about laboratory quality measeurements was added to 2.4 section |
Line 122-125 |
Section 2.6: The use of proportions is fine for the main discussion of this article. But physical scientists want and need data of actual concentrations. These data MUST be presented in at least an SI, there does seem to be some of these data in the results. Nonetheless, please add a table to the SI with the median and standard deviation for each group along with the porportion data. This MUST happen to improve the impact of this study and support science by contributing to the greater data pool on ASGM human health data |
It was included a supplementary table with the median mercury levels for some variables according to the exposition group. Median is used because of the assimetric distribution of mercury levels in each matrix and so IR is the appropriate deviation measure relating the median. |
Table Supplementary 2 introduced in the results section text lines 170-171 |
Lines 137-139: This comparison doesn't make sense. How many older women completed primary school and how many younger women complete secondar school. Why highlight different things from different age groups? |
The sentence was re-written to present a clear idea about education |
Lines 140-142 |
Lines 152-159: Is this the actual concentration values I have been asking for? These data must have units associated with them or they mean very little. Again, I would recommend adding a table with all the concentration data in the SI. This supports open science and also will allow other scientists to potential use such data in for future studies, such as meta studies looking at combining data on concentations in women at all ASGM sites globally. |
Table Supplementary 2 was added with median mercury levels medians |
Table Supplementary 2 introduced in the results section text lines 170-171 |
Section 4.1: The new section 4.1 is all background information and should be moved to the introduction. The current section 4.2 should be the start of the discussion. |
This section was added according to reviewer#1 suggestion and so it is part of the response for the initial comments, approved by this reviewer. |
|
Lines 231-240: This is again all repeated information from the introduction. Delete it. Get to the point of the discussion... |
The paragraph was deleted |
Lines 230-239 |
Line 237: "CAN UNDERGO METHYLATION!!!" Not all mercury gets methylated!!! |
This sentence was deleted according to the previous comment |
Line 237 |
Line 247: What is "the earlier"...I have no idea. This is not an expression used to describe an earlier point. "The former" maybe?! |
The word was replaced |
Line 240 |
Lines 253-254: "On the other side" of what? Again not an English expression. Just say, "exposure is likely exacerbated by the long average residence time for inhabitants in the Mojana area..." |
The sentence was modified |
Lines 246-247 |
Lines 284-286: This contradicts what the authors discussed in the previous section that higher education (as assumed socioeconomic status) means women eat more fish...so which is it?! |
A possible explanation is previously stated in the former paragraph. Despite socioeconomic status, in coastal cities fish consumption culd be higher, relating occupation habits in the region like fishing. |
Lnes 257-259 |
Lines 292-298: I feel like this whole paragraph is just over explanation of the health concerns described in the introduction...again. Mercury is bad for pregnant womens health. We understand this, as the authors have described it at length!! Delete this paragraph and get onto describing how this relates to your data (the next paragraph). Line 297: What is "metilmercury?" |
The paragraph was deleted |
Line 287 |
Lines 301-304: Again we know these health effects!!! you've said it over and over and over again...too much repitition. Delete. |
The sentence was deleted |
Line 289 |
Lines 323-325: Not therefore...please write: "some of those foods (e.g. Brasil Nuts) have been reported to have a mediating effect on mercury in the human body." |
The sentence was modified |
Lines 307-309 |
The manuscript was revised and adjusted by a native English speaker, in order to improve the style and the language overall.
Reviewer 3 Report
I congratulate the authors for the new approach of their proposal and the prompt response to the suggested comments.
Author Response
Reviewer #3
Reviewer comment |
Adjustment done |
Answered in |
I congratulate the authors for the new approach of their proposal and the prompt response to the suggested comments. |
The reviewer does not have additional suggestions for the authors |
None |
The manuscript was revised and adjusted by a native English speaker, in order to improve the style and the language overall.